# Interphase Cytogenetic Analysis of G0 Lymphocytes Exposed to α-Particles, C-Ions, and Protons Reveals their Enhanced Effectiveness for Localized Chromosome Shattering—A Critical Risk for Chromothripsis

**DOI:** 10.3390/cancers12092336

**Published:** 2020-08-19

**Authors:** Antonio Pantelias, Demetre Zafiropoulos, Roberto Cherubini, Lucia Sarchiapone, Viviana De Nadal, Gabriel E. Pantelias, Alexandros G. Georgakilas, Georgia I. Terzoudi

**Affiliations:** 1Laboratory of Health Physics, Radiobiology & Cytogenetics, Institute of Nuclear & Radiological Sciences & Technology, Energy & Safety, National Centre for Scientific Research “Demokritos”, 15341 Athens, Greece; gabriel@ipta.demokritos.gr; 2DNA Damage Laboratory, Physics Department, School of Mathematical and Physical Sciences, National Technical University of Athens, 15780 Zografou, Greece; alexg@mail.ntua.gr; 3Legnaro National Laboratories, Italian Institute of Nuclear Physics, 35020 Legnaro (Padova), Italy; Demetre.Zafiropoulos@lnl.infn.it (D.Z.); Roberto.Cherubini@lnl.infn.it (R.C.); lucia.sarchiapone@lnl.infn.it (L.S.); viviana.denadal@lnl.infn.it (V.D.N.)

**Keywords:** high-LET particle radiation, protons, α-particles, C-ions, premature chromosome condensation (PCC), PCC assay, chromatin dynamics, chromothripsis, localized chromosome shattering, chromothripsis-like chromosomal rearrangements, RBE values, fingerprint of exposure, radiation oncology, space radiation protection

## Abstract

For precision cancer radiotherapy, high linear energy transfer (LET) particle irradiation offers a substantial advantage over photon-based irradiation. In contrast to the sparse deposition of low-density energy by χ- or γ-rays, particle irradiation causes focal DNA damage through high-density energy deposition along the particle tracks. This is characterized by the formation of multiple damage sites, comprising localized clustered patterns of DNA single- and double-strand breaks as well as base damage. These clustered DNA lesions are key determinants of the enhanced relative biological effectiveness (RBE) of energetic nuclei. However, the search for a fingerprint of particle exposure remains open, while the mechanisms underlying the induction of chromothripsis-like chromosomal rearrangements by high-LET radiation (resembling chromothripsis in tumors) await to be elucidated. In this work, we investigate the transformation of clustered DNA lesions into chromosome fragmentation, as indicated by the induction and post-irradiation repair of chromosomal damage under the dynamics of premature chromosome condensation in G0 human lymphocytes. Specifically, this study provides, for the first time, experimental evidence that particle irradiation induces localized shattering of targeted chromosome domains. Yields of chromosome fragments and shattered domains are compared with those generated by γ-rays; and the RBE values obtained are up to 28.6 for α-particles (92 keV/μm), 10.5 for C-ions (295 keV/μm), and 4.9 for protons (28.5 keV/μm). Furthermore, we test the hypothesis that particle radiation-induced persistent clustered DNA lesions and chromatin decompaction at damage sites evolve into localized chromosome shattering by subsequent chromatin condensation in a single catastrophic event—posing a critical risk for random rejoining, chromothripsis, and carcinogenesis. Consistent with this hypothesis, our results highlight the potential use of shattered chromosome domains as a fingerprint of high-LET exposure, while conforming to the new model we propose for the mechanistic origin of chromothripsis-like rearrangements.

## 1. Introduction

For precision cancer radiotherapy purposes, the biological effects of energetic nuclei with high linear energy transfer (LET) have received growing attention lately, particularly regarding the mechanisms underlying their increased relative biological effectiveness (RBE) and potential risk for induction of secondary malignancies. Interestingly, despite their significant therapeutic benefits, particle irradiation has recently been reported to induce chromothripsis-like complex chromosomal alterations, similar to those generated by the phenomenon of chromothripsis in tumors [1,2]. Rather than by a stepwise accumulation of subsequent alterations, chromothripsis is a mutational process in which large stretches of chromosomes undergo massive but localized shattering and random rearrangements in response to a one-step catastrophic event [3,4,5,6]. By provoking inaccurate rejoining of chromosome fragments, this phenomenon leads to a new genome configuration and the formation of complex chromosomal alterations that may cause carcinogenesis [7,8,9,10,11,12], by amplification of oncogenes, for instance [13]. Therefore, full understanding of the processes underlying chromosome shattering and the formation of chromothripsis-like complex chromosomal alterations is an important step towards the clarification of the increased biological effectiveness and long-term health risk of high-LET particle irradiation. In this respect, we have recently provided experimental evidence supporting that localized chromosome shattering in micronuclei (MN) is triggered in a single catastrophic event by the dynamics of premature chromosome condensation (PCC) in asynchronous micronucleated cells [14]. Consequently, we consider it of interest to examine whether the dynamics of chromatin condensation during the cell cycle can transform persistent DNA and chromatin alterations into breaks, thereby explaining the increased efficacy of particle irradiation for killing cancerous cells and the formation of chromothripsis-like chromosomal alterations.

Towards this aim, we analyze chromosomal damage directly in interphase cells following the traversal of particle radiation through the nucleus, since chromosome alterations are correlated to both early (e.g., cell killing) and late effects (e.g., carcinogenesis) [15]. Low- and high-LET studies carried out in vitro and in vivo demonstrate that the types of chromosome aberrations observed, and the biological impact of an exposure depends on the radiation quality and energy deposited [16]. As a result of their low-density energy deposition, χ- and γ-rays induce sparsely distributed damage, causing mostly indirect DNA lesions via increased oxidative stress to the interphase chromosomes that occupy localized domains of the nucleus [17,18,19]. Among the DNA lesions, double-strand breaks (DSBs) in the G0/G1 phase are the most dangerous, since there is no complementary strand available (like in the G2 phase) that could serve as a repair template [20,21,22,23]. In contrast to low-LET radiation, particle irradiation deposits high-density energy that is expected to induce direct focal DNA damage in chromosome domains along the particle tracks, leading to effective cell killing and increased tumorigenicity. In fact, accumulated evidence suggests that the increased RBE of high-LET radiation compared with photons is driven by the formation of complex DNA lesions [12,24,25,26], defined as DNA damage containing both DSBs and single-strand breaks (SSBs), as well as base damage within 1–2 helical turns. This is also referred to as clustered DNA damage and, together with clustered DSBs, defined as multiple DSBs, are generally accepted as the key lesions that determine the effectiveness of high-LET radiation [23,24,25,26,27,28,29,30,31,32]. However, their consequences at the chromosomal level and, particularly their role in the formation of chromothripsis-like chromosomal alterations are not clearly understood. This is mainly because the spectrum of DNA lesions induced by the traversal of high-LET particles through nuclei has proven very complex and difficult to study. Indeed, the experimental validation of the induction of clustered DNA damage and the comprehension of the repair mechanisms involved have not been easy tasks [23,31]. In particular, the yields reported on chromosome aberrations and their complexity are time dependent due to repair effects, cell cycle delays, and the removal of non-surviving and apoptotic cells from the sample. This fact complicates the interpretation of the results obtained by means of the conventional analysis of irradiated cells at metaphase. Irradiation, especially with high-LET particles, retards the entry of damaged cells into mitosis and, therefore, one major problem in the analysis of heavy-ion induced aberrations is the cell cycle delay and G2-block, which depend not only on LET and dose, but also on the cell type [33,34,35,36].

In the present work, we investigate the impact of clustered DNA lesions, as indicated by the induction and post-irradiation repair of chromosomal damage, directly in interphase chromosomes. For this purpose, we use a clearly detectable cytogenetic endpoint of exposure in order to obtain reliable RBE values of different radiation qualities, as well as to investigate the mechanisms underlying the induction of chromosome shattering and the formation of chromothripsis-like chromosomal alterations. Towards this goal, the fusion PCC assay is employed to visualize and analyze chromosome fragmentation directly in G0 human lymphocytes, without the requirement of exposed cells entering into mitosis [37,38]. Specifically, lymphocytes isolated from whole blood were exposed to various doses (up to 6 Gy) of α-particles (4.70 MeV, 92 keV/μm), accelerated C-ions (56.5 MeV, 295 keV/μm), and protons (2.2 MeV, 28.5 keV/μm). The yields of chromosome fragmentation obtained for induction and post-irradiation repair (up to 24 h) were compared with those obtained for γ-rays, in order to derive RBE values. Furthermore, we tested the hypothesis that clustered DNA lesions and persistent chromatin decompaction induced by high-LET irradiation at the damage sites along the particle tracks, can subsequently evolve into localized chromosome shattering if chromatin condensation occurs. The detection and quantification of such localized shattering of chromosome domains induced by each type of radiation quality was enabled by means of the PCC assay and a rigorous interphase cytogenetic assessment. The observed cytogenetic effect has the potential to serve as a fingerprint of high-LET exposure and instructs our proposal of a new model of the mechanistic origin of chromothripsis-like complex chromosomal rearrangements following particle irradiation.

## 2. Results

### 2.1. Interphase Cytogenetic Analysis of G0 Lymphocytes by Means of the PCC Assay is a Promising Tool to Study the Mechanisms Underlying the Biological Effectiveness of Particle Irradiation

Cells irradiated with α-particles, accelerated C-ions, and protons, face a drastic alteration of their cell cycle kinetics and increased difficulties to reach mitosis. To overcome this problem, the PCC assay offers a unique tool to study induction and repair of radiation-induced chromosome aberrations directly in G0 lymphocytes, without the requirement of exposed cells entering into mitosis. A representative image of non-irradiated peripheral blood G0 lymphocytes exhibiting 46 prematurely condensed chromosomes (PCCs) is shown in Figure 1. The stable number of 46 PCCs in non-irradiated blood samples can be considered as a clearly detectable interphase cytogenetic endpoint. Indeed, it allows the detection and quantification of radiation-induced DNA lesions, as reflected at the level of interphase chromosomes by means of excess (over 46) PCC fragments. In the present study, the yields obtained for the different radiation qualities were used to derive RBE values and to investigate the mechanisms underlying their distinct effectiveness. As a result of low-density energy deposition, γ-rays induce chromosome fragmentation in interphase lymphocytes with a mostly random distribution among the chromosome domains in the nucleus, as shown in Figure 2A.

In contrast to γ-rays, the traversal of particle radiations through the nucleus deposits high-density energy that mainly induces—even at high doses—direct intense localized DNA lesions. These DNA lesions can be transformed into chromosome fragments only in the domains along the particle tracks, leaving thus intact the non-targeted chromosomes, as shown in the lower part of Figure 2B, for α-particles, as well as in Figure 3 for C-ions and protons. To analyze and quantify the impact of particle radiation-induced clustered DNA lesions at the chromosome level, we define a “localized shattered chromosome domain” as the fragmentation of an interphase chromosome into five or more clearly detectable fragments in close proximity. Such localized shattered chromosome domains, as shown by arrows in Figure 3A,B for C-ions and protons, respectively, may be used as a fingerprint of exposure to particle radiations. Furthermore, this observed cytogenetic endpoint may be easily quantified and may as well play a potential role in the elucidation of the mechanisms underlying differences in effectiveness among different radiation qualities, as explained in the paragraphs below and in the Discussion section.

The PCC analysis enables the follow-up examination of the observed chromosome shattering at progressing times after irradiation. In the case of low-LET radiation, such as γ-rays, a significant reduction in the number of excess PCC breaks takes place. This reduction of chromosome fragmentation reflects the processing of the underpinning subsets of DNA damage in interphase G0 lymphocytes during the time between exposure and analysis. Regarding particle irradiation, a reduction in the number of excess PCC breaks may also be observed, as shown in Figure 4A for 24 h post-irradiation repair at 37 °C following a 6 Gy exposure to α-particles. However, a high percentage of the irradiated lymphocytes under the same conditions exhibit increased yields of excess PCC fragments and shattered chromosome domains even at 24 h repair time, as shown by arrows in Figure 4B. This observation suggests the presence of persistent particle radiation-induced clustered DNA lesions and chromatin alterations, even at 24 h post-irradiation repair time.

### 2.2. RBE Values for Different Radiation Qualities Can Be Obtained Using Chromosome Fragmentation Analysis Directly in Interphase G0 Lymphocyte PCCs

RBE values were obtained using the PCC assay for the assessment and quantification of radiation-induced chromosomal aberrations directly in human peripheral blood G0 lymphocytes, following exposure to different radiation qualities. Compared with γ-rays, a significant increase in damage induction and subsequent yields of excess PCCs per nucleus was obtained for doses up to 6 Gy of α-particles, C-ions, and protons, as depicted in Figure 5. Based on our analysis, the RBE values obtained for induction of chromosomal damage were calculated to be 4.1 for α-particles, 2.6 for C-ions, and 2.1 for protons.

Our studies also revealed differences in the repair kinetics of radiation-induced chromosomal aberrations in G0 lymphocytes for the different radiation qualities used. The yields of excess PCCs following repair were quantified by the residual un-rejoined fragments at various post-irradiation repair times up to 24 h at 37 °C, as presented in Figure 6. Increased post-repair RBE values were obtained, when compared with those obtained for the initial induction of chromosomal damage. Specifically, post-repair RBE values at 24 h were found to be 10.7 for α-particles, 5.4 for C-ions, and 3.9 for protons.

### 2.3. Shattered Chromosome Domains are a Fingerprint of Exposure to High-LET Particle Radiation and Their Yield Depends on Dose and Radiation Quality

An additional aim of our experimental design was to search for a specific fingerprint of exposure to high-LET particle radiation, since this important issue remains open and the data are controversial [15]. Towards this goal, we exploited our observation that focal deposition of high-density energy by particle irradiation can shatter a targeted chromosome domain along the particle tracks into several (five or more) clearly detectable fragments in close proximity (Figure 3). Indeed, such shattered chromosome domains are very frequent in G0 lymphocyte PCCs following exposure to high-LET particle radiations, compared with γ-rays. For instance, when G0 lymphocytes are exposed even to only 1 Gy of α-particles, three shattered chromosome domains can be scored in the PCC spread shown in Figure 7A, but none in the case of exposure to 1 Gy of γ-rays, as shown in Figure 7B. Therefore, such localized shattering of a targeted chromosome domain visualized in G0 lymphocyte PCCs may be considered as a fingerprint of exposure to high-LET particle radiation. The yields of shattered chromosome domains per nucleus, following exposure to various doses up to 6 Gy for the different radiation qualities used, are shown in Figure 8. Linear dose–response relationships were obtained, with RBE values of 14.3 for α-particles, 7.5 for C-ions, and 4.9 for protons.

### 2.4. Persistent Shattered Chromosome Domains May Explain Differences in Biological Effectiveness among Different Radiation Qualities and the Induction of Chromothripsis-Like Rearrangements

The transformation of particle irradiation-induced clustered DNA lesions into the observed localized shattering of chromosome domains is revealed in our experiments by means of the premature chromosome condensation dynamics. Yet, this transformation can also take place by means of the cell cycle dependent chromatin condensation dynamics when cells proceed to G2/M phase transition, assuming that the lesions induced by particle radiation in the targeted chromosome domains are persistent. In order to investigate this assumption, G0 lymphocytes were irradiated with 6 Gy of α-particles, C-ions, protons, and γ-rays. The yields of shattered chromosome domains were obtained, either immediately after exposure or at 24 h post-irradiation repair time, as shown in Figure 9. Particle radiation-induced persistent lesions in the chromosome domains of a nucleus remain higher for α-particles with an RBE value of 28.6, followed by C-ions with 10.5, and protons with 4. All the RBE values calculated in the present work are summarized in Table 1. Persistent lesions are thus of importance, since chromatin dynamics during G2/M phase may transform them into localized chromosome shattering, a hallmark of chromothripsis. Moreover, random rejoining of shattered chromosomes may evolve into chromothripsis-like rearrangements, as we describe in the Discussion section.

## 3. Discussion

### 3.1. Can Clustered DNA Lesions Alone Account for the Formation of Complex Chromosomal Aberrations and the Increased Relative Biological Effectiveness of Particle Irradiation?

While it is commonly accepted that clustered DNA damage is characteristic of high-LET radiation, the mechanisms through which it causes complex and, particularly, chromothripsis-like alterations [1,2] similar to those generated by the phenomenon of chromothripsis in tumors, have not yet been clarified. This is mainly because the scale of clustered DNA damage is in the order of 10–30 bp, i.e., <3–4 nm, whereas the scale of chromosomal rearrangements is in the order of >1000 bp, i.e., >50–100 nm distance. To explain this discrepancy between clustered DNA lesions and chromosomal rearrangements in terms of scale, the possibility of mis-rejoining two DSBs belonging to distinct loci must be considered [39]. Indeed, this difference may be resolved by the existence of clustered DSBs in close proximity along the tracks of high-LET particle radiation. Specifically, 3D-structured illumination microscopy revealed the formation of clustered DSBs within γH2AX foci signals in C-ion-irradiated G2 phase cells [40]. While clustered DNA damage is typically defined by the presence of additional lesions in the immediate vicinity of the DSBs, clustered DSBs or multiple DSBs represent a further level of overall damage complexity. As a result, clustered DSBs likely add a substantially higher accident risk to any repair process attempt [21,23]. Therefore, this form of damage may underpin the increased efficacy of high-LET radiation, since clustered DSBs are more challenging to repair and have a larger probability of lethality [12,24,25,26]. The spectrum of clustered DNA lesions induced by the traversal of high-LET particles through nuclei has proven very challenging and difficult to study, when considering interactions only at the DNA level and not at the chromosome level, where the dynamics of chromatin conformation changes during the cell cycle come into play. However, elucidating the link between clustered DNA lesions and formation of complex chromosome rearrangements is crucial for both particle radiotherapy and space radiation protection.

### 3.2. Chromosome Aberration Analysis of G0 Lymphocyte PCCs Enables the Assessment of DNA Damage without the Requirement of Irradiated Cells Entering into Metaphase

In the present study, our experimental strategy focused on the impact of clustered DNA lesions on chromosome fragmentation under the dynamics of chromatin organization changes in interphase cells. As shown in Figure 1, the fusion PCC assay offers a clearly detectable cytogenetic endpoint of exposure based on chromosome fragmentation analysis in G0 lymphocyte PCCs. The presence of 46 prematurely condensed chromosomes in non-irradiated lymphocytes represents the normal human genome and is, thus, a clearly stable number without variability among healthy blood donors. As a result of energy deposition by ionizing radiation, the induced clustered DNA lesions can be transformed into interphase chromosomal fragmentation with a random and homogeneous distribution following low-LET exposure (Figure 2A) [41,42]. However, following high-LET particle irradiation, the focal deposition of high-density energy induces clustered DSBs and affects chromatin at their sites only in the chromosome domains along the particle tracks [23,43]. As a result, localized chromosome shattering occurs under the dynamics of chromatin condensation—even at high doses—as shown in Figure 2B for α-particles, in Figure 3A for C-ions, and in Figure 3B for protons. Analysis of PCCs at progressing times after irradiation up to 24 h can show a reduction in the number of excess PCC breaks. This reduction of chromosome fragmentation reflects the processing of the underpinning subsets of DNA damage and formation of rings and translocations during the time between exposure and analysis in interphase G0 lymphocytes (Figure 4A) [44]. However, a high percentage of the irradiated lymphocytes under the same conditions exhibit the presence of localized chromosome shattering even at 24 h post-irradiation repair time (Figure 4B and Figure 9), plausibly due to persistent clustered DNA lesions and chromatin alterations.

### 3.3. Reliable RBE Values for Particle Radiations Can Be Established Using Fragmentation of Interphase Chromosomes as a Biological Endpoint

To obtain dose–response relationships for different radiation qualities, the yield of radiation-induced chromosome fragments in interphase G0 lymphocytes can be expressed as excess (over 46) PCCs (shown in Figure 2). The results obtained demonstrate that excess PCCs increase linearly with radiation dose and that high-LET radiation generates a higher level of excess PCCs than low-LET γ-rays (Figure 5). An examination of the reduction in excess PCCs up to 24 h post-irradiation reveals that the fragments decrease with time but are at a significantly higher level following high-LET radiation (Figure 6). Furthermore, a quantification of shattered chromosome domains shows as well that their frequency increases linearly with radiation dose (Figure 8), whereas at 24 h post-irradiation repair time persistent shattered chromosome domains remain higher for α-particles, followed by C-ions and protons, being the least for γ-rays (Figure 9). When compared with γ-rays, the RBE values obtained using the above biological endpoints, range from 4.1 to 28.6 for α-particles, 2.6 to 10.5 for C-ions, and 2.1 to 4.9 for protons, as presented in Table 1.

Top of FormBottom of FormEarlier studies using fusion-induced PCC in G1 cells from mammalian cell lines reported RBE values ranging only from 1 to 2 for high-LET α-particles [45,46] or swift heavy-ions [47,48,49,50,51]. However, recent experiments at the Heavy Ion Medical Accelerator (HIMAC) in Japan using G0/G1 phase of normal human fibroblasts report values as high as 30 at low doses [52,53]. For protons, an RBE of 1.1 is used clinically, although this depends on various physical and biological factors and there is an ongoing debate about its accuracy (reviewed in [54]). Based on the endpoints examined in the present work, we calculated higher RBE values (2.1–4.9) for protons, which could be the result of the high LET values obtained (28.5 keV/μm) when using low energy protons (2.2 MeV incident energy). Overall, the increased RBE values obtained in our repair experiments (Table 1) can be of interest, since the intrinsic radiosensitivity of normal or tumor human cells often correlates with the level of residual breaks [48,55,56].

### 3.4. Localized Chromosome Shattering Induced by Energetic Nuclei May be Used as a Fingerprint of Exposure

When using conventional cytogenetics for chromosome aberration analysis at metaphase, high RBE values are obtained if analyzed in the first post-irradiation metaphases, but they are only close to 1 if the analysis takes place in the progeny of irradiated cells. Such data may suggest that most of heavy ion induced chromosome aberrations are non-transmissible to the progeny cells. Consequently, this observation could translate into diminishing late effects and, therefore, could be considered a significant advantage of particle radiotherapy. However, such possibility should be validated, particularly in the light of recent reports that show formation of chromothripsis-like alterations related to potential late effects of particle irradiation [1,2]. Indeed, these reports provide evidence that particle radiation can induce chromothripsis-like complex chromosomal alterations similar to those generated by the phenomenon of chromothripsis in tumors. These new data could have a profound impact on RBE for potential late effects of energetic nuclei and, especially for initiation of carcinogenesis by specific chromosome rearrangements. In this respect, it is crucial to understand how chromothripsis-like complex chromosomal alterations can be formed following high-LET exposure and, to what extent, the localized chromosome shattering within the nucleus can be initiated by particle irradiation. Therefore, the identification of a fingerprint of particle radiation exposure, particularly regarding the late effects and carcinogenesis, is important.

When G0 lymphocytes are exposed even to only 1 Gy of α-particles, three shattered chromosome domains can be scored in the PCC spread shown in Figure 7A, but none can be scored in the case of exposure to 1 Gy of γ-rays, as shown in Figure 7B. These results suggest that high-LET particle radiations are more efficient biologically because they generate more shattered chromosome domains along their tracks in the nucleus. Indeed, the observed localized shattering of interphase chromosomes, with high risk for random rejoining, is probably the precursor of complex intra-chromosomal rearrangements that are recognized as characteristic events of particle irradiation [57,58]. Furthermore, our observation that α-particles, accelerated C-ions, and protons have a dose-dependent enhanced effectiveness to induce localized chromosome shattering (Figure 8), a hallmark of chromothripsis, and may contribute towards the identification of a fingerprint of exposure that could be related to an increased risk of secondary malignancies. Specifically, clustered DNA damage and clusters of DSBs, which are induced by the focal deposition of high-density energy in chromosome domains along the particle tracks, can be reflected and visualized in G0 lymphocyte PCCs as localized shattered chromosome domains. Such localized chromosome shattering is characterized by the presence of multiple closely spaced fragments within individual chromosomes and could be considered as a fingerprint of exposure (Figure 3, Figure 4B and Figure 7A). The RBE values established using such a fingerprint as biological endpoint are significantly higher than those obtained for excess PCCs/nucleus (Table 1). Altogether, our results suggest that localized chromosome shattering can be used to elucidate the mechanistic origin of the differences in biological effectiveness obtained experimentally for the various radiation modalities used (Figure 8 and Figure 9).

### 3.5. Our Model: Clustered DNA Lesions are Transformed through Chromatin Condensation into Localized Chromosome Shattering and, via Random Rejoining, Evolve into Chromothripsis-Like Rearrangements

At the DNA level, particle irradiation is expected to induce a variety of complex DNA lesions, challenging the DNA repair enzymatic machinery (reviewed in [59]). A shifting of the DNA damage response (DDR) system towards less-accurate non-homologous end joining (NHEJ) repair pathways for DSBs and all other neighboring DNA lesions has been suggested [60]. It is a general current consensus that the fidelity of repair depends on the complexity of the lesion, with clustered DSBs being more difficult to repair than isolated breaks [61]. Complex DSBs, either formed directly by irradiation or by the processing of non-DSB clustered lesions, are expected to be processed by slow kinetics or left unrepaired and cause cell death or pass mitosis. In these surviving cells, large deletions, translocations, and chromosomal aberrations have been detected (reviewed in [62]). Nevertheless, the exact mechanism underlying the transformation of clustered DNA lesions into such chromosomal alterations is not yet clearly understood. Based on our previously published evidence on the dynamics of chromatin condensation changes in transforming DNA lesions with chromatin decompaction at their sites into visual breaks [41,42,63], we propose here that high-LET radiation induces not only clustered DNA lesions, but also physical perturbation of the chromatin organization and persistent chromatin decompaction at the sites of clustered DNA lesions. This dual action of energetic nuclei on DNA and chromatin is a key characteristic of high-LET radiation and can translate into localized chromosome shattering in a one-off event, by the dynamics of chromatin condensation during the cell cycle. The fact that shattering of chromosome domains can be observed immediately after irradiation by means of premature chromosome condensation (Figure 8), together with their relative persistence (Figure 9), suggest that the dynamics of chromatin condensation can initiate the single destructive event needed for the phenomenon of chromothripsis to occur. Indeed, localized chromosome shattering in neighboring chromosome domains along the particle tracks followed by random rejoining may evolve into chromothripsis-like complex chromosomal rearrangements.

According to our hypothesis, when clustered DNA lesions and their associated chromatin decompaction are induced by particle irradiation, e.g., in the S phase, their conversion into localized chromosome shattering may also take place by means of cell cycle dependent chromatin condensation dynamics as cells proceed to the G2/M-phase transition. Therefore, the critical parameter for radiation-induced chromothripsis is not the dose itself but the radiation quality and the potential of high-LET particle irradiation to induce persistent clustered DNA lesions and chromatin decompaction at their sites. Direct experimental evidence reinforcing this view was provided recently by Timm et al. [43]. These authors demonstrated experimentally that clustered DNA damage concentrated in particle trajectories causes persistent rearrangements in chromatin architecture, which may affect the structural and functional organization of cell nuclei. In fact, they demonstrated that chromatin decompaction and remodeling during repair of clustered DNA damage fails to restore the original nucleosomal organization at damage sites. On the contrary, after low-LET irradiation, the induced single DSBs throughout the nucleus in euchromatin and heterochromatin were efficiently repaired without damage-associated large-scale remodeling of chromatin. Their results suggest, therefore, that the impact of low-LET radiations on chromatin is not persistent. This difference in response at the chromatin level, together with the induction of clustered DNA lesions, constitute the high-LET dual action that we consider in our model to explain why particle radiations are more prone to induce chromothripsis-like rearrangements.

Chromothripsis-like chromosomal rearrangement could be as well generated following particle radiation-induced localized chromosome shattering through the formation of MN in the progeny of irradiated cells [4,14,64]. Since particle radiation-induced clustered DNA lesions and chromatin decompaction at their sites are more persistent than those induced by low-LET radiation [43], they have an increased capacity to proceed to G2/M transition and undergo chromatin condensation. As a result, chromosome fragmentation will take place leading to aberrant cell mitosis and formation of micronucleated cells via asymmetrical cell division. When main nuclei in micronucleated cells enter mitosis, premature chromosome condensation in MN provokes shattering of the chromosomes entrapped inside micronuclei, if they are still undergoing DNA replication and thus have maximum chromatin decompaction at their replication sites. Under these conditions, chromatin condensation dynamics exert mechanical stress causing DNA replication forks to collapse into DSBs, leading to localized chromosome shattering in a single catastrophic event that may be followed by random rejoining and subsequently evolve into chromothripsis-like chromosomal rearrangements in the progeny cells, as we have recently proposed [14].

## 4. Materials and Methods

### 4.1. Cell Cultures and Preparation of PCC-Inducer Mitotic CHO Cells

Chinese Hamster Ovary (CHO) cells were grown in McCoy’s 5A (Biochrom, Berlin, Germany) culture medium supplemented with 10% FBS, 1% l-glutamine, and 1% antibiotics (Penicillin, Streptomycin), and incubated at 37 °C in a humidified atmosphere with 5% CO_2_. CHO cultures were maintained as exponentially growing monolayer cultures in 75 cm^2^ plastic flasks at an initial density of 4 × 10^5^ cells/flask. Colcemid (Gibco) at a final concentration of 0.1 μg/mL was added to CHO cultures for 4 h and the accumulated mitotic cells were harvested by selective detachment. Once a sufficient number of mitotic cells had been obtained, they were used as supplier of mitotic promoting factors (MPF) to induce PCC in human lymphocytes. The mitotic CHO cells harvested from one 75 cm^2^ flask were used for 2–3 fusions.

### 4.2. Lymphocyte Isolation from Human Peripheral Blood

Peripheral blood samples in heparinized tubes were obtained from healthy male and female donors. Lymphocytes were isolated from whole blood using Biocoll separating solution (Biochrom). The blood samples diluted 1:2 in RPMI-1640 without FBS were carefully layered on top of equal amounts of Biocoll in 14 mL test tubes and centrifuged at 400× *g* for 20 min. Collected lymphocytes from each tube were washed with 10 mL culture medium (RPMI-1640 supplemented with 10% FBS, 1% glutamine, and antibiotics), centrifuged at 250× *g* for 10 min, and kept in culture medium before irradiation with different radiation qualities. Lymphocytes isolated from 1 mL of blood were used for each experimental point.

### 4.3. Irradiation and Sample Preparation

In the present work, lymphocytes isolated from whole peripheral blood were exposed to a gamma irradiator, a source of α-particles, accelerated C-ions, and to a proton beam. For γ-rays exposure, irradiation of lymphocytes was carried out in vitro using a Co-60 Gamma Cell 220 irradiator (Atomic Energy of Canada Ltd., Ottawa, Canada) at room temperature with 1.3 MeV photons and LET at 0.2 keV/μm with a dose rate of 20 cGy/min. Lymphocyte suspensions in culture medium were exposed for different times in order to deliver doses ranging from 0 to 6 Gy. Following irradiation, lymphocytes were either processed immediately for their fusion and PCC induction (a procedure that permits approximately a repair time of 1 h) or allowed to repair at 37 °C for different times up to 24 h. Subsequently, the samples were processed for cell fusion, PCC induction, and preparation of microscope slides in order to analyze the PCC spreads.

For exposure to α-particles, a Curium-244 alpha source (Isotope Producers Laboratories, CA, USA) was used with particle energy 4.70 MeV at the cell surface entrance and LET at 92 keV/μm. For C-ions exposure, accelerated Carbon-12 ions at 56.5 MeV with LET at 295 keV/μm was applied. For exposure to protons, a proton beam with an incident energy of 2.2 MeV and LET of 28.5 keV/μm was used. For the particle irradiation, lymphocytes were exposed as a monolayer in a special cube. The density of lymphocyte suspension was adjusted using an inverted microscope so that lymphocytes were exposed as a monolayer in contact to each other without gaps.

For α-particles (perpendicular beam), 20 μL of dense lymphocyte suspension were diluted in 0.5 mL of medium, it was loaded into the cube and allowed to sediment onto a mylar surface of 6 μm thickness and 13 mm diameter before sample irradiation. For protons and accelerated C-ions (horizontal beam), 20 μL of dense lymphocyte suspension was sandwiched between two mylar surfaces, enabling thus their exposure to the horizontal beams. The irradiation of samples was carried out at the Legnaro Lab accelerators and the homogeneous exposure of the targeted cells in the entire mylar surface by the radiation beam was ensured by means of appropriate testing. The dose-rate for α-particles was 0.22 Gy/min, whereas for proton irradiation was 1 Gy/min and for 295 keV/um carbon-ion irradiation was 2 Gy/min. The energies of particle and ion irradiations, as well as the experimental set-up, were chosen in a way to always guarantee the so-called “track segment conditions” (or to be very close to these) in order to obtain the correct evaluation of the deposited energy in the cell and then of the dose [65,66]. All particle energy values, and corresponding LET refer to incident energy at the cell entrance. Considering that in our experiments the targeted cells were lymphocyte monolayers, the particles traversed the cells retaining considerable energy, so the Bragg peak was not totally contained in the exposed sample, even though the biological endpoints proposed in this work were found to be sensitive enough to distinguish the effectiveness of the different radiation qualities used.

### 4.4. Cell Fusion-Mediated Induction of Premature Chromosome Condensation in Lymphocytes

Cell fusion and PCC induction were performed using 45% polyethylene glycol (PEG, P5402, Sigma-Aldrich, Darmstadt, Germany) in serum-free RPMI-1640 medium. Lymphocytes and mitotic CHO cells were mixed in serum-free RPMI-1640 medium in a 14 mL round-bottom culture tube in the presence of colcemid as originally described [67] with some modifications. After centrifugation at 1000 rpm (100× *g*) for 8 min, the supernatant was discarded without disturbing the cell pellet, keeping the tubes always inverted in a test tube rack on a paper towel to drain the pellet from excess liquid. While holding the tubes in an inverted position, 0.15 mL of PEG was injected forcefully against the cell pellet using a micropipette and, immediately after, the tube was turned in an upright position and held for about 1 min. Subsequently, 1.5 mL of PBS was slowly added to the tube with gentle shaking and the cell suspension was centrifuged at 1000 rpm for 8 min. The supernatant was discarded, and the cell pellet was suspended gently in 0.7 mL RPMI-1640 complete growth medium containing PHA and colcemid. After 75 min at 37 °C, cell fusion/PCC induction was completed. Cells were then treated with hypotonic KCl (0.075 M) and fixed with three changes of methanol: glacial acetic acid (*v*/*v* 3:1).

### 4.5. Cytogenetic Analysis, Scoring Criteria, Statistical Analysis

The chromosome spreads were prepared by the standard air-drying technique and slides were stained using 3% Giemsa in buffered solution for PCC analysis. The PCC fragments per cell characterized as “Excess PCCs/Cell” (i.e., in excess of 46 PCCs) were scored for damage induction or post-irradiation repair points using light microscopy. The analysis of PCC spreads was greatly facilitated by an image analysis system (Ikaros, MetaSystems, Germany). Detection and quantification of the impact of radiation-induced clustered DNA lesions (i.e., clustered DNA damage and clustered DSBs) on interphase chromosomes, in terms of localized chromosome shattering, was made through the visualization and subsequent analysis of the 46 chromosome domains in G0 lymphocyte PCC spreads. For this purpose, we defined “shattered chromosome domains/nucleus” as the yields per nucleus of shattered interphase chromosomes containing five or more clearly detectable fragments in close proximity to each other (e.g., Figure 3, Figure 4B and Figure 7A). For each experimental point, at least 50 cells (G0 lymphocyte PCC spreads) were analyzed and the experimental results shown in Figure 5, Figure 6, Figure 8, and Figure 9, represent mean values ± SD based on three independent experiments for α-particles, C-ions, and γ-rays; and two independent experiments for protons. Statistical significance was determined by means of unpaired *t*-tests, corrected for multiple comparisons using the Holm–Sidak method with alpha = 0.05. Each dose was analyzed individually, without assuming a consistent SD. Asterisks indicate statistical significance; * *p* ≤ 0.05, ** *p* ≤ 0.01, *** *p* ≤ 0.001.

## 5. Conclusions

By means of a clearly detectable biological endpoint, we obtained reliable RBE values for α-particles, C-ions, and protons, and studied the mechanisms underlying the efficacy of particle irradiation to induce localized chromosome shattering, a hallmark of chromothripsis. Specifically, based on chromosome fragmentation analysis of G0 lymphocyte PCCs, our test system reflects the impact of radiation-induced clustered DNA lesions on induction and post-irradiation repair in interphase chromosomes. The results provide the first direct experimental evidence that high-LET particle radiations have an increased effectiveness for localized chromosome shattering in domains along the particle track. This specific effect is shown to be a fingerprint of exposure, which can improve our understanding and unravel the differences in biological effectiveness exhibited by various radiation qualities. It points as well to our proposal of a new model for the mechanisms underlying the formation of critical complex chromosome alterations.

Indeed, the potential of particle irradiation to induce persistent lesions at the level of DNA as well as of chromatin, in neighboring chromosome domains along the particle tracks, may be a key determinant of the formation of chromothripsis-like chromosomal rearrangements. Such dual action of particle radiation may lead to localized chromosome shattering under the dynamics of chromatin condensation, which may be followed either by random rejoining of chromosome fragments, or aberrant mitosis and MN formation. In both cases, chromothripsis-like rearrangements similar to those caused by chromothripsis in tumors may be generated with a potential impact on long-term health risks. High-LET particle radiation is more likely, therefore, to cause complex focal genomic changes leading to a higher level of genomic instability. To examine this possibility, additional insights into the fate of the localized shattered chromosomes could be obtained for different radiation qualities by combining the PCC assay with the Fluorescence in Situ Hybridization (FISH) technique, as we have already demonstrated for low-LET radiation [44,68]. Overall, our results are of importance to radiation oncology and space radiation protection, since the induction of complex and chromothripsis-like alterations by particle radiation may generate adverse effects and increased risk of secondary malignancies.

## Figures and Tables

**Figure 1 cancers-12-02336-f001:**
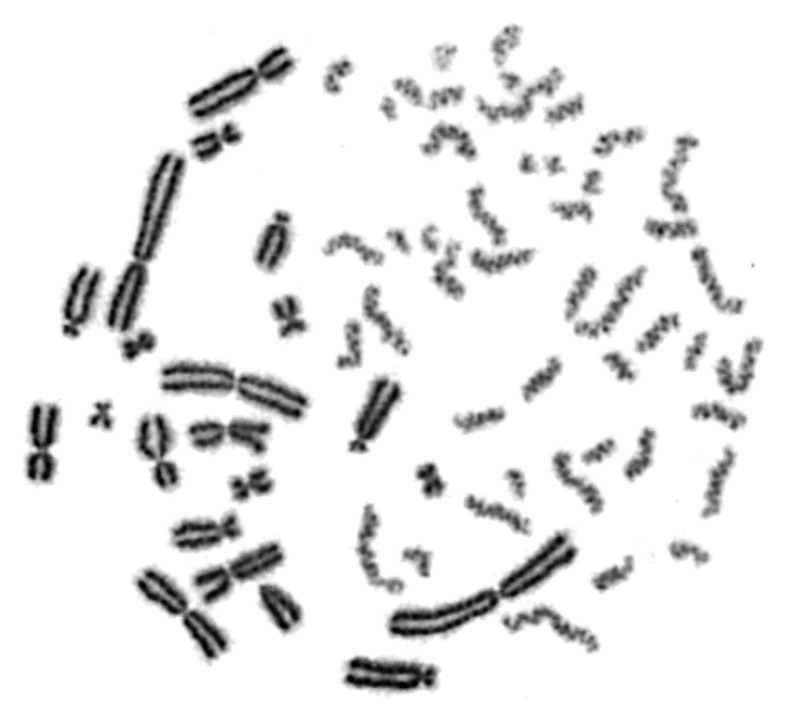
Representative image of a non-irradiated peripheral blood G0 lymphocyte exhibiting 46 lightly stained single chromatid prematurely condensed chromosomes (PCCs) obtained by fusion with a mitotic Chinese Hamster Ovary (CHO) cell. The stable number of 46 PCCs in non-irradiated peripheral blood lymphocytes from healthy donors represents a clearly detectable interphase cytogenetic endpoint. In fact, it allows quantification of radiation-induced DNA lesions, as reflected at the level of interphase chromosomes by means of excess (over 46) PCC fragments.

**Figure 2 cancers-12-02336-f002:**
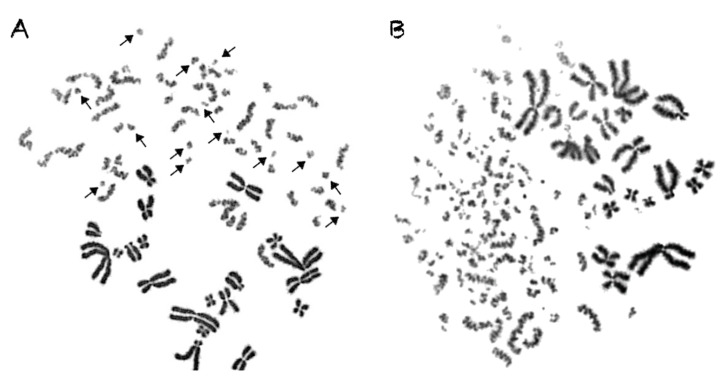
Irradiated peripheral blood G0 lymphocytes with 4 Gy of γ-rays or 6 Gy of α-particles. (**A**) The impact of low-density energy deposition by 4 Gy of γ-rays translates into chromosome fragmentation in interphase lymphocyte PCCs with a sparse distribution among the chromosome domains of the nucleus, as shown by arrows. Fourteen excess (over 46) PCC fragments can be scored. (**B**) In contrast to γ-rays, even high-density energy deposited by 6 Gy of high-LET α-particle radiation induces direct focal clustered DNA lesions. These DNA lesions can be transformed into localized chromosome shattering in the domains along the particle track, leaving thus intact the non-targeted chromosomes, as shown in the lower part of the PCCs in this panel. One hundred excess (over 46) PCC fragments can be scored.

**Figure 3 cancers-12-02336-f003:**
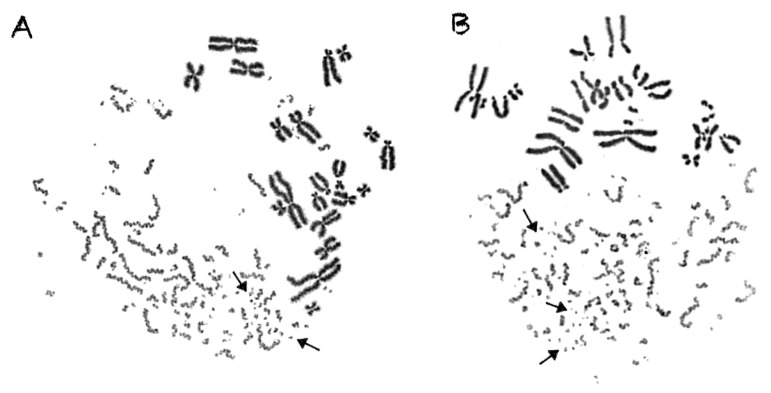
High-LET particle irradiation induces localized chromosome shattering, as revealed in G0 lymphocyte PCC spreads. For analysis and quantification purposes regarding the impact of particle radiation-induced clustered DNA lesions at the level of interphase chromosomes, we define localized shattered chromosome domains as the fragmentation of an interphase chromosome into five or more clearly detectable fragments in close proximity to each other. Examples are shown by arrows for a G0 lymphocyte spread obtained following exposure to 6 Gy of accelerated C-ions (**A**) and 4 Gy of protons (**B**). It is apparent that the non-targeted chromosome domains remain intact.

**Figure 4 cancers-12-02336-f004:**
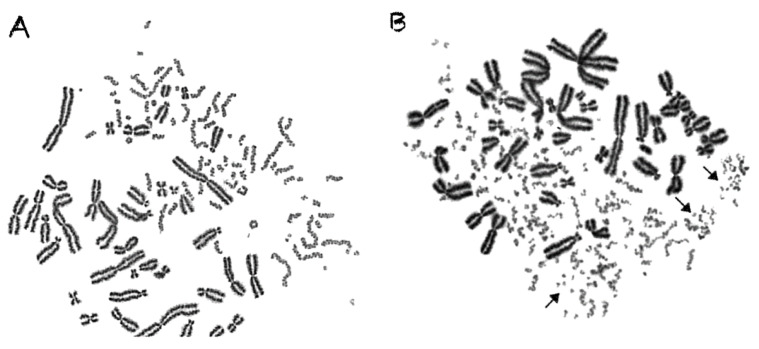
Following exposure to different radiation qualities, persistent shattered chromosome domains can be visualized and quantified in G0 lymphocyte PCC spreads. (**A**) Ring formation and a reduction in the number of excess PCC breaks may be observed at 24 h post-irradiation repair time at 37 °C, following exposure to 6 Gy of α-particles. (**B**) Under the same conditions, however, most of the irradiated lymphocytes still exhibit a high number of excess PCC fragments and shattered chromosome domains after 24 h of repair, as shown by arrows.

**Figure 5 cancers-12-02336-f005:**
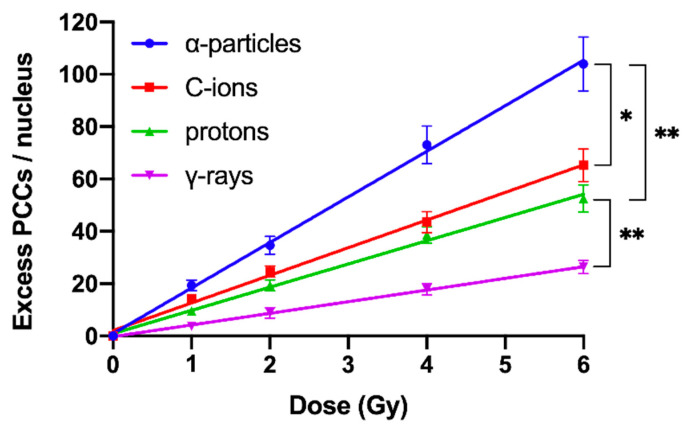
Dose–response curves for the initial DNA lesions induced by different radiation qualities as reflected in G0 lymphocyte PCCs and constructed by means of excess PCC fragments per nucleus at various doses up to 6 Gy of α-particles, C-ions, protons, and γ-rays. Based on this cytogenetic endpoint of initial induction of radiation damage, the relative biological effectiveness (RBE) values obtained were 4.1 for α-particles, 2.6 for C-ions, and 2.1 for protons. (Mean ± SD based on three independent experiments; *n* ≥ 50 cells analyzed per experimental point; * *p* ≤ 0.05, ** *p* ≤ 0.01).

**Figure 6 cancers-12-02336-f006:**
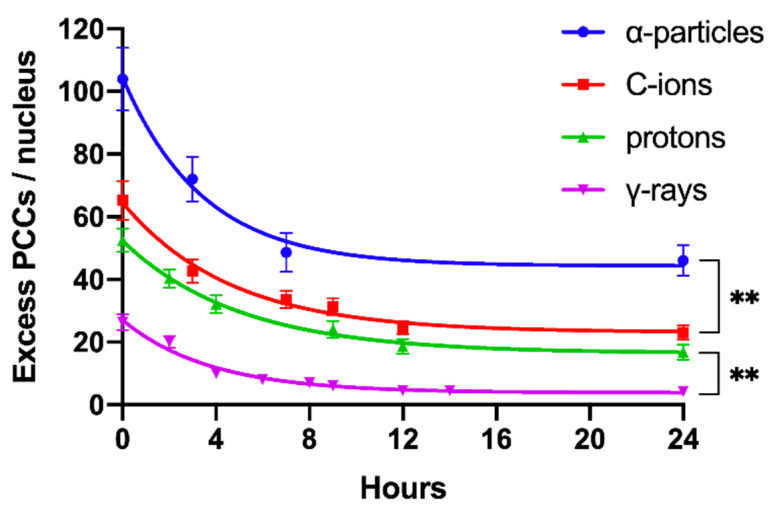
Repair kinetics of the initial DNA lesions induced by 6 Gy of different radiation qualities as reflected in G0 lymphocyte PCCs and constructed by means of excess PCC fragments per nucleus at various repair times up to 24 h. Based on this cytogenetic endpoint of residual lesions at 24 h repair time, the derived RBE values were 10.7 for α-particles, 5.4 for C-ions and 3.9 for protons. (Mean ± SD based on three independent experiments; *n* ≥ 50 cells analyzed per experimental point; ** *p* ≤ 0.01).

**Figure 7 cancers-12-02336-f007:**
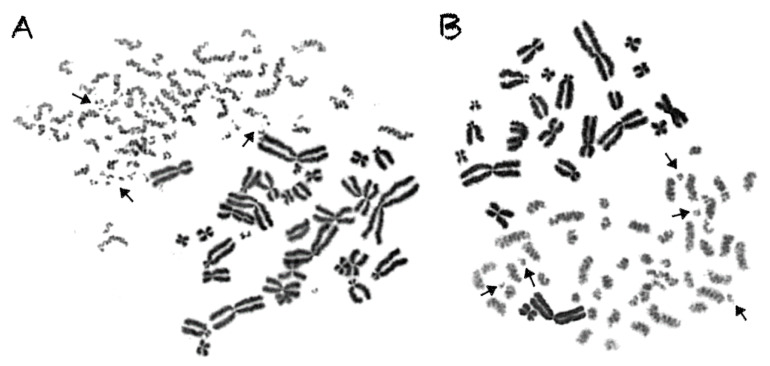
Shattered chromosome domains are very frequent in G0 lymphocyte PCCs following exposure to high-LET particle radiations, compared with γ-rays. (**A**) Exposure to even 1 Gy of α-particles can result in three localized shattered chromosome domains, as shown by arrows. (**B**) In contrast, following 1 Gy of γ-rays, only single randomly distributed chromosome fragments could be visualized. Therefore, such localized shattering of targeted chromosome domains may be considered as a fingerprint of exposure to high-LET particle radiation.

**Figure 8 cancers-12-02336-f008:**
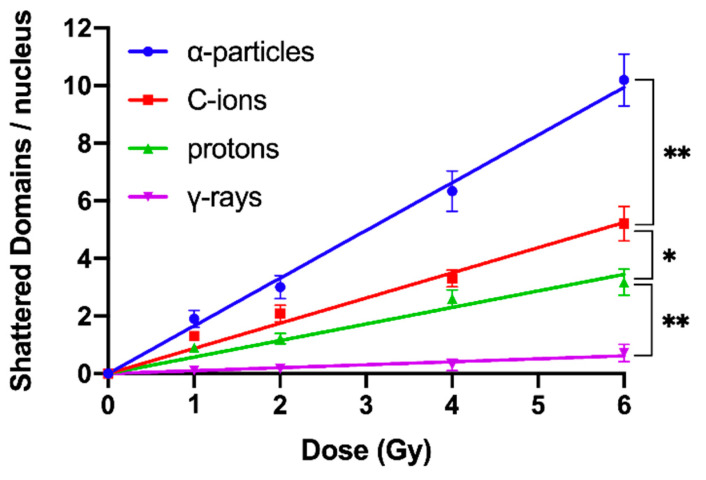
Dose–response curves for the yields of shattered chromosome domains per nucleus, following exposure to various doses up to 6 Gy of the different radiation qualities. Linear dose–response relationships were obtained with most effective being the α-particles, followed by C-ions, protons, and γ-rays. Based on this cytogenetic endpoint of the initial induction of clustered DNA lesions and formation of shattered chromosome domains, the RBE values obtained were 14.3 for α-particles, 7.5 for C-ions, and 4.9 for protons. (Mean ± SD based on three independent experiments; *n* ≥ 50 cells analyzed per experimental point; * *p* ≤ 0.05, ** *p* ≤ 0.01).

**Figure 9 cancers-12-02336-f009:**
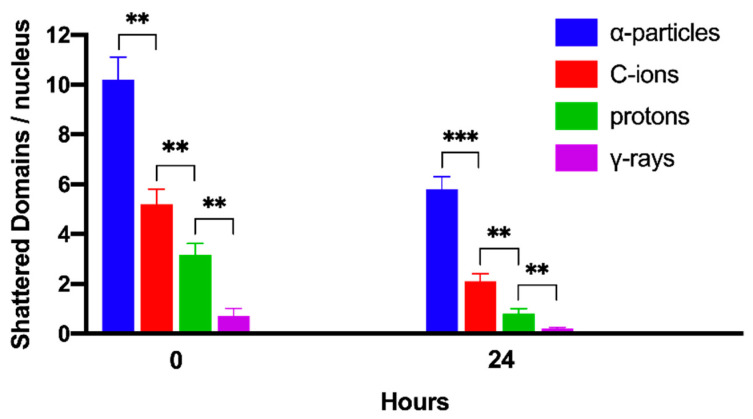
Persistent shattered chromosome domains at 24 h repair time, following exposure to different radiation qualities. G0 lymphocytes were irradiated with 6 Gy of α-particles, C-ions, protons or γ-rays and the yields of shattered chromosome domains were obtained, either immediately after exposure (0 h) or at 24 h post-irradiation repair time. Particle radiation-induced persistent shattered chromosome domains of the nucleus remain higher for α-particles, followed by C-ions and protons, being the least for γ-rays. Based on this cytogenetic endpoint of persistent shattered chromosome domains after 24 h, the RBE values obtained were 28.6 for α-particles, 10.5 for C-ions, and 4 for protons. (Mean ± SD based on three independent experiments; *n* ≥ 50 cells analyzed per experimental point; ** *p* ≤ 0.01, *** *p* ≤ 0.001).

**Table 1 cancers-12-02336-t001:** Summary of all the RBE Values Compared with γ-rays.

Radiation Quality	RBE Excess PCCs/Nucleus		RBE Shattered Domains/Nucleus	
	0 h	24 h	0 h	24 h
α-particles	4.1	10.7	14.3	28.6
C-ions	2.6	5.4	7.5	10.5
protons	2.1	3.9	4.9	4

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
