# Peer review of "Interphase Cytogenetic Analysis of G0 Lymphocytes Exposed to α-Particles, C-Ions, and Protons Reveals their Enhanced Effectiveness for Localized Chromosome Shattering—A Critical Risk for Chromothripsis"

_cancers, 2020, doi:10.3390/cancers12092336_

Round 1
Reviewer 1 Report
This manuscript follows a previous paper published by the same authors in Cancers in 2019 in which they demonstrated that premature chromosome condensation dynamics during asynchronous mitosis in micronucleated or multinucleated cells are an important determinant of chromosome shattering and may underlie the mechanistic origin of chromothripsis.
In this article, the same type of experiment is repeated in response to different ions (α-particles, carbon ions, protons). This new article appears to be less successful with a certain number of errors or missing information. In my opinion, it is too premature for publication in Cancers journal for the following reasons:
- The experimental results are based on a calculation of the relative biological effectiveness(RBE) which is false. They calculate it as ” the yield of chromosomal damage obtained at 6 Gy for two types of radiation” wherea RBE is defined as “the ratio of the doses required by two radiations to cause the same level of effect”;
- The doses used are variable without any justification: fig 1: 4 Gy gamma-ray and 6 Gy alpha particles, fig 3: no dose, fig 5: 1 Gy gamma-ray and 1 Gy alpha particles;
- For the interpretation of data, it would be important to have the calculation of the fluence and the mean number of particles per nucleus;
- The number of cells analyzed is not specified for each experiment;
- The correspondence between 295 keV/µm and 56,5 MeV seems unlikely for the carbon ion irradiation;
- The authors claim that “the chromosome shattering is localized along the ion tracks” and that the "non-targeted chromosomes domains remain intact" but they do not provide any experimental argument to demonstrate it. In the same way they make assumptions about the decompaction of chromatin without experimental results and about chromothripsis without CGH-array confirmation;
- Fig 7, there is no match between the legend text (three-localized shattered chromosomes domains) and the value of 2 in the figure;
- The number of lymphocyte donors is not specified. Is there an impact of the individual radiosensitivity on the results observed? This could create a bias in the difference observed between the different beam types. Are the manipulations repeated on different donors?
- The impact of the ion type on the results observed is not discussed despite the 6 pages of discussion;
- The 6 page discussion is far too long.
Reviewer 2 Report
This well designed and innovative investigation gives experimental evidence that high LET irradiation induces localized fragmentation of targeted chromosome domains. The post repair RBEs for chromosome fragmentation were found for 2.2 MeV protons, 5 MeV alphas, and 56 MeV carbon ions when compared to gamma irradiation. The methods employed excess premature chromosome condensation (PCC) fragments per nucleus and the yields of shattered chromosome domains per nucleus. The authors tested the hypothesis that clustered DNA lesions and persistent chromatin decompaction induced by high-LET irradiation, results in chromosome shattering on chromatin condensation. Their novel results provided evidence that lead to a new model for the mechanistic origin of chromothripsis-like chromosomal alterations via high LET radiation. A significant result of this work is that it shows that alpha particles are more effective than carbon ions or proton in shattered chromosome yields that are formed initially and even more effective after repair. Overall this is an excellent work that presents an improved model for chromosome damage by heavy ion irradiation.
Comments:
- The authors find that RBE values for α-particles (92 keV/μm), for C-ions (295 keV/μm) and for protons (28.5 keV/μm). These ions will penetrate to differing depths into the cell cultures. In section 4.3 some discussion is given of the experimental set up but it is not clear if these beams penetrate through the cell cultures or whether the Bragg Peak is contained in the sample. This needs more explanation since track structure and energy deposition changes along the track.
Reviewer 3 Report
The authors of this manuscript investigate the phenomenon of chromosome shattering, related to the process of chromothripsis, in lymphocytes after exposure to ionising radiation of differing linear energy transfer (LET) that generate increased amounts and complexity of DNA damage. This was achieved using an assay for measurement of prematurely condensed chromosomes (PCCs) in response to low-LET γ-rays in comparison to high-LET α-particles, carbon ions and protons. It is proposed that 46 PCC fragments are present in non-irradiated lymphocytes, and that PCCs above this baseline increase linearly with radiation dose. High-LET radiation also generates a higher level of excess PCCs than low-LET γ-rays. An examination of the reduction in excess PCCs up to 24 h post-irradiation revealed that these decrease with time, but are at a significantly higher level following high-LET radiation. Finally, a quantification of shattered domains, similar to PCCs, show that their frequency increase linearly with radiation dose.
In general, the manuscript is written and presented well, and describes an important piece of work. I therefore have only a few relatively minor comments that should be addressed.
Specific comments:
- There is no statistical analysis of the data particularly in Figures 5, 6, 8 and 9, which is a limitation, that I assume is due to the lack of appropriate number of experiments (minimum of 3). If this isn’t possible, then it may be best to indicate which dataset acquired from each of the radiation modalities is from either 2 or 3 experiments, to put these data in the correct context.
- Images show chromosome fragmentation after 4 Gy γ-rays (Figure 4A) and 6 Gy α-particles (Figure 4B), but no dose is provided for carbon ions and protons (Figure 3A/B). I would actually suggest to provide images at the equivalent radiation dose (either 4 or 6 Gy) of all the modalities, which is then reflective of the excess PCCs quantified in Figure 5. It would also be helpful if images reflecting the PCC data in Figure 6 (e.g. at 24 h) and shattered domains in Figure 8 (e.g. 6 Gy) are provided for γ-rays, carbon ions, protons and α-particles as Supplementary Information, not just the ones shown as examples in the respective figures. This would allow a direct visual comparison.
- The authors infer, due to the increase in LET, that chromosome shattering is as a consequence of increases in clustered DNA damage but do not show this directly. It would have been nice to correlate the linear dose-dependent increases in excess PCCs/shattered domains with the different radiation modalities, with measurements of this DNA damage (e.g. immunofluorescence, PFGE or comet assays). This though is just a comment, and is not absolutely essential for the current manuscript.
- Clinically, an RBE of 1.1 for protons is used, although this depends on various factors including dose, dose rate, LET (clustered DNA damage), repair capacity of the cell and biological end-point measured (an appropriate citation to include is PMID:31284432). However, the end-points examined here for protons provide an RBE ranging from 2-4.5, which appear relatively high. The methodology states that low energy protons (2.2 MeV incident energy) were utilised that may describe the high-LET values obtained (28.5 keV/µm), and thus higher RBE. Nevertheless, some brief comment on this should be added to the Discussion. Furthermore, details of the positioning of the cells relative to the Bragg peak (spread-out Bragg peak?) should be added as well as dose rates used in the Methods. Similarly, dose rates for carbon ions and α-particles should be stated.
- A stable number of 46 PCC fragments in non-irradiated blood samples is implied, is this a precise number or are there variabilities between samples?
- The Discussion is quite extensive, and particularly Sections 3.3 and 3.4 are very overlapping with the Results section. It is suggested to shorten and compact this part of the manuscript to make it more concise.
- There are several grammatical errors throughout that require correction. For example:-
Line 60: “Consequently, we consider of interest to..” should be changed to “Consequently, we considered it of interest to..”.
Line 316: “This fact makes problematic the…”, change to “This fact creates problems with the interpretation…”.
Line 406: “shuttered”, change to “shattered”.
Reviewer 4 Report
The authors have used premature chromosome condensation to compare interphase chromosome damage induced by low LET (gamma rays) with that induced by three different high LET radiation sources in human lymphocytes. The data presented are straightforward and reveal that high LET radiation tends to cause multiple relatively closely spaced breaks within individual chromosomes, although the data would have been strengthened by FISH analysis to confirm that the fragments arise from the same chromosomes. The authors also established RBE values for the phenomenon. The paper is clearly written but I have several concerns.
- The major issue is that the limited, albeit clear, data presented in the paper do not warrant such a long and wide ranging Discussion. A shorter and more focused Discussion would improve the manuscript.
- In places the claims about the results are somewhat overstated. For example, the authors state “In the present study, we provide experimental evidence indicating that particle irradiation induced clustered DNA damage result in shattering of chromosome domains” (lines 445-6), but this is an inference since no direct measurements were taken of clustered lesions. Since three different sources of high LET radiation were employed, the authors should perhaps show a direct relationship between the types and levels of clustered lesions induced by each of these three high LET radiations and the respective number of chromosome fragments.
- The repair data shown in Figure 6 are interesting because the rates of decrease of excess chromosome fragments appear to be fairly similar for all the radiation types, including low and high LET. While it is clear that more lesions persist after 12 h, this could be due to lesions that are intrinsically unrepairable or that the repair system(s) is exhausted. Following repair after exposure to a second round of radiation at 12 h might be able to resolve this question.
- Minor points:
(a) The word “shattered” is written “shuttered” in several places.
(b) What is the commercial source of the Curium-244?
Round 2
Reviewer 4 Report
A few minor errors:
line 326 - plausibly
line 419 - cell-cycle dependent
line 446 - may be followed
Author Response
"Please see the attachment."
